# Development and evaluation of the digital-screen exposure questionnaire (DSEQ) for young children

**Nimran Kaur**[1], **Madhu Gupta**[1]*, **Tanvi Kiran**[1], **Prabhjot Malhi**[2], **Sandeep Grover**[3]

**1** Department of Community Medicine and School of Public Health, Postgraduate Institute of Medical Education and Research (PGIMER), Chandigarh, Union Territory, India, **2** Department of Pediatrics, PGIMER, Chandigarh, Union Territory, India, **3** Department of Psychiatry, PGIMER, Chandigarh, Union Territory, India

* madhugupta21@gmail.com

## Abstract

### Background

Over the last three decades, the accessibility and usage of mobile devices have increased among young children. This study's objective was to develop a validated caregiver-reported digital-screen exposure questionnaire (DSEQ) for children aged 2–5 years.

### Methods

DSEQ was developed in five phases. Phase 1, a draft questionnaire was developed by reviewing the literature on existing tools (n = 2) from 2009–2017. Phase 2, face-to-face interviews with primary caregivers (n = 30) were conducted in a tertiary-care-hospital for acculturation. Nine experts assessed the face and content validity of the draft Hindi and English questionnaire. Phase 3, a pilot study conducted among randomly selected families (n = 40) to evaluate the feasibility of DSEQ in field settings. Phase 4, test-retest reliability was done among 30 primary caregivers selected randomly in another urban cluster. Phase 5, the internal consistency of DSEQ was checked by conducting a cross-sectional study among randomly selected 400 primary caregivers in Chandigarh, North India. IBM SPSS Statistics for Macintosh, version 25.0, was for data management and analysis.

### Results

A valid DSEQ with 86 items under five domains, including sociodemographic, screen-time exposure and home media environment, level of physical activity, media-related behaviors, and parental perceptions was developed. The pilot study showed that it was feasible to use the DSEQ in the field. DSEQ was reliable with kappa value ranging from 0.52 to 1.0, and intra-class coefficient of 0.62–0.99 (p<0.05). A strong internal consistency was observed for three domains including, screen-time exposure and home media environment (Cronbach's alpha of 0.82), media-related behaviors (Cronbach's alpha of 0.74) and physical activity (Cronbach's alpha 0.73).

**Data Availability Statement:** All the relevant data are within the paper and its Supporting information files.

**Funding:** NK is supported by Indian Council of Medical Research (3/1/3/Next-100/JRF-2015/HRD-

SS/03/10451/135), New Delhi for doing her PhD. https://www.icmr.gov.in MG applied for Intramural project funds from PGIMER have also funded the project, Chandigarh–160012 [71/2-Edu-16/92, Dated 08/01/2018]. https://pgimer.edu.in/PGIMER_PORTAL/PGIMERPORTAL/home.jsp Neither of the funders played any role in the study design, data collection, analysis, decision to publish or preparation of the manuscript.

**Competing interests:** The authors declare no potential conflicts of interest.

## Conclusions

The developed DSEQ has good face and content validity and acceptable evidence of internal consistency and test-retest reliability. The DSEQ can be used for measuring digital screen exposure and its correlates among children aged 2 to 5 years.

## Introduction

Caregivers regulate the digital ecosystem within which their children grow up [1]. A child learns new skills by impersonating their parents or family members regarding media-use [2]. Developing an understanding of how caregivers influence their child's motivation is principally vital as children grow up and become autonomous of their recreational time [3]. Over the last three decades, the accessibility and usage of mobile devices have increased among young children. The prevalence of excessive digital-screen exposure (DSE) in under-five children varies from 21% to 98% in middle-income countries [1].

A Thai study in infants and toddlers reported that 1-year-olds watched television (TV) for 1.23 (SD = 1.42) hours per day, which increased to 1.69 (SD = 1.56) hours per day when they turned two years [4]. It raises a pertinent research question on the effects of unregulated DSE on young users' overall health. Children's passive media-viewing behavior adversely affects their health and social outcomes [5], delay motor skills, cognitive development, and language development [6,7]. This type of behavior might lead to non-communicable diseases later in life [8]. Therefore, it is imperative to correct these excessive digital-media-focused sedentary lifestyles at an early age [9].

Despite DSE's high prevalence and numerous ill effects, standardized measurement tools do not evaluate the DSE accurately [10]. Using questionnaires with appropriate validity and reliability levels reduces errors and measures the desired outcome [3]. One of the major hurdles is the accurate estimation of the DSE's duration and correlates of DSE, as available instruments do not evaluate the cumulative exposure from all sources (like TV, computers, smartphones, etc.) comprehensively [1]. Previous studies have used the tools that have measured DSE primarily from TV [4]. In contrast, children use many other gadgets like handheld videogames, tablets, iPad®, etc. [1].

A review by Kaur et al. (2019) suggested that there is a need to assess the DSE's prevalence among under-five children in the developing countries including India to understand how they use digital-media devices [1]. However, it was reported that only limited validated tools existed to measure DSE [11–13]. Among these tools one was in Chinese and the other two in English language [1]. The existing tools were developed for older age-groups (3–6 years [13], 5–15 years [12], and above 18 years [11]). There was no tool that comprehensively measured DSE and its correlates among toddlers including intrapersonal, interpersonal, immediate media micro-environment and macro-environment associates. Measuring a young child's DSE and its correlates with a validated tool is important as DSE might be associated with screen-based sedentary behaviours affecting health outcomes like emotional behaviours [6,10], sleep problems [14] in the early childhood or development of non-communicable diseases during the adolescense [9] and adulthood [4]. As the media-exposure access and usage patterns are different among different populations so the country-specific data needs to be generated.

This study's objectives were to develop a comprehensive DSE questionnaire (DSEQ) for children aged 2–5 years to assess the cumulative DSE (including all types of digital-media

devices used by the child routinely) and its correlates (including parental digital-media behaviors, parental digital-media literacy, home-media environment, media-content watched by the child, physical activity of the child, activities performed by the child, etc.). This study's findings might help the researchers to assess the burden of DSE, especially in the developing countries, and impact of interventions to reduce the DSE using a validated tool, and help in formulating the age-specific policies to regulate the DSE among preschoolers.

## Methods

The DSEQ was developed in four phases, as shown in Fig 1.

### Phase 1: Item development

Published articles from January 2009 to June 2017 were searched in Pubmed, Scopus, Embase, Clinical Key, and Google Scholar using Medical Subject Heading (MeSH) words to develop items in a preliminary DSEQ. MeSH words are given in published review by the lead author [1]. This phase was conducted from January to June 2017.

The items related to the child's level of physical activity on a weekday and weekend were classified at five progressive levels, i.e., stationary no movement, stationary with limb or trunk movement, slow-, medium-, or fast-paced activity [15], was also incorporated.

### Phase 2: Assessing face and content validity

Initially, four rounds of the face-to-face interviews with 30 primary-caregivers (6–8 per round) of children aged 2–5 years, attending the Pediatric outpatient department of a tertiary-care-hospital in Chandigarh were conducted in a separate room for acculturation of DSEQ. The eligibility criteria for the participants were written informed consent to participate in the study, and education up to middle school. The lead author interviewed the primary-caregivers individually using the preliminary DSEQ (English version) in July 2017. In each interview, the parents were asked about their understanding of digital-media usage in children and any missing areas or concerns in the questionnaire. After each round of face-to-face interviews, the draft DSEQ was modified in consultation with three experts (authors themselves), one each from the specialty of Psychology, Psychiatry, and Public Health. These experts by rewording, addition, deletion, and rearrangement of the questions modified the DSEQ for acculturation. New and modified questions were used to interview a new set of primary caregivers in each

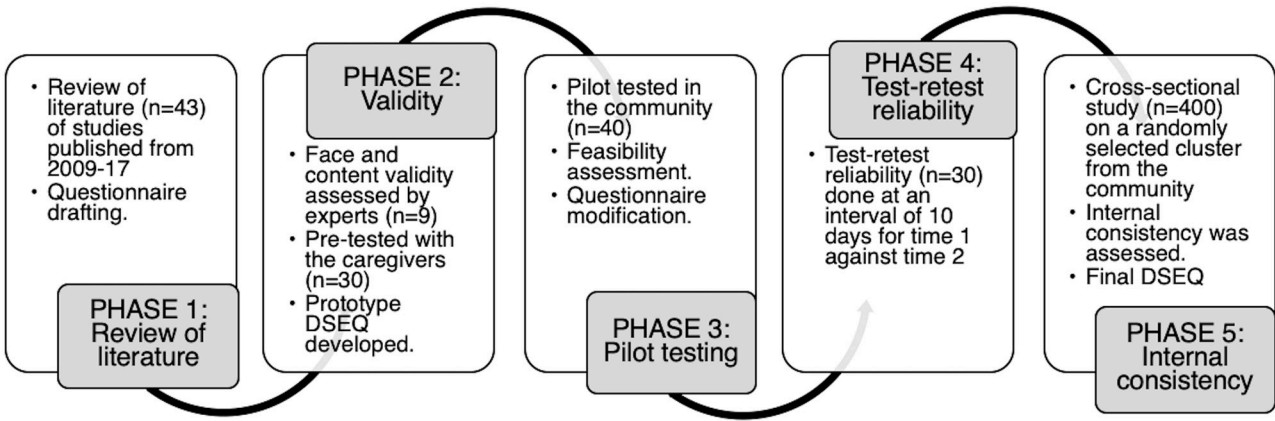

**Fig 1. Phases of designing the digital-screen exposure questionnaire (DSEQ).**

round, until all possible questions/responses were included. The rearrangement, addition or removal of an item was checked by two experts independently, and finalized by a third one in case of any disagreement. The questionnaire was sequentially translated into the Hindi language by bilingual health professionals (n = 3) using standard translation-back translation methodology [16].

Later, the bilingual DSEQ was given to 25 different experts from the fields of Pediatrics, Public Health, Community Medicine, Psychology, Psychiatry, and Pediatric Neurology. They had at least five years of experience and a doctor's degree (Doctor of Medicine/Ph.D.). These chosen experts were approached through the mail or personal meetings to assess the content (structural construct, utility/ futility of questions, arrangement/ flow, sections/ sub-sections, and domain-wise distribution) and face validity (overall sequence/ language and appropriateness). The experts were given the conceptual definitions of the underlying constructs, the study's objectives, and a Likert-type feedback proforma. (S1 Table). Each question's score in the feedback proforma ranged from 1 to 4 where, 1 = 'not at all', 2 = 'minimally', 3 = 'to a certain extent', and 4 = 'to a large extent'. The score on the length of the questionnaire ranged from 0 to 4, where 0 = 'very short', 1 = 'short', 2 = 'adequate', 3 = 'long', and 4 = 'very long'. Qualitative comments on the questions were also asked. Of the 25 experts contacted, nine experts reverted. The questionnaire was further modified, as per the comments of the experts.

**Phase 2: Data management and analysis.** Microsoft Excel version 2019 was used for generating random numbers. The DSEQ versions were saved in Microsoft Word version 2019 and circulated among the experts for their inputs.

## Phase 3: Feasibility assessment

A pilot study was conducted to assess how feasible was to use the prototype DSEQ in the home environment of the families in the community. This study was conducted in the urban settings in Sahib Ajit Singh Nagar, district Mohali, Punjab, India, in August 2017. All families having children aged 2–5 years, at least one TV/smartphone/digital-media gadget in their homes and consented to be a part of the study were included. Children previously diagnosed with long-term, or chronic or congenital illnesses were excluded. A multi-stage simple random sampling was done to select 40 random families from the study area. In the first stage, a catchment population area of one auxiliary-nurse-midwife in Phase I, Sahib Ajit Singh Nagar was randomly selected. In the second stage, the families with children between 2–5 years were randomly selected from the catchment area by computer-generated random numbers. The selected families were then contacted by the lead author by visiting their homes to conduct the face to face interviews with the primary caregivers of the children using the prototype DSEQ (developed in phase 2), at a time convenient to them. The information regarding the children was proxy-reported by their primary caregivers.

**Phase 3: Data management and analysis.** IBM SPSS Statistics for Macintosh, version 25.0, and Microsoft Excel version 2019 were used for data management and analysis. The proportion of children with excessive DSE (more than one hour per day) as per the American Academy of Pediatrics guidelines, 2016, was estimated [9]. DSE's average duration from all the gadgets was calculated separately for weekdays (Monday to Friday) and weekend days (Saturday and Sunday). The weighted average of DSE per day (minutes) was calculated by using the formula = [(Weekday DSE time in minutes x 5) + (Weekend DSE time in minutes x 2)]/ 7.

## Phase 4: Test-retest reliability

The test-retest reliability was done among randomly selected primary caregivers of children aged 2–5 years in September 2017, to assess the psychometric properties of the DSEQ. This

stage was conducted in urban settings in Phase III, Sahib Ajit Singh Nagar, district Mohali, Punjab, India. Sampling frame was families with children aged 2–5 years residing in the study area. All families having children aged 2–5 years, at least one TV/smartphone/digital-media gadget in their homes and consented to be a part of the study were included. Children previously diagnosed with long-term, or chronic or congenital illnesses were excluded. A multi-stage simple random sampling technique was used to select 30 families from the study area. In the first stage, a catchment population area of one auxiliary-nurse-midwife in Phase III, Sahib Ajit Singh Nagar, Punjab was randomly selected. In the second stage, the families with children between 2–5 years were randomly selected from the catchment area by computer-generated random numbers. The chosen families were contacted at their homes. The lead author interviewed the primary caregivers by face-to-face interview using the prototype DSEQ (modified in Phase 3) at time convenient to them. The information regarding the children was proxy-reported by their primary caregivers.

The data's repeatability was observed on two occasions (time one and time two), ten days apart. At time one, the lead author interviewed the willing primary-caregivers in their homes. At time two, ten days later, these participants were again approached by the lead author to assess the differences in their responses.

**Phase 4: Data management and analysis.** Microsoft Excel version 2019 was used for data entry. The IBM SPSS Statistics for Mac, Version 25.0. Armonk, NY (2017) was used for measuring Cohen's Kappa and Intraclass Correlation Coefficient (ICC) values. The results were estimated at a 95% confidence interval. We had measured Cohen's Kappa values and ICC for three domains of DSEQ including, screen-time exposure and home media environment with 16 items (domain 2), level of physical activity with 2 items (domain 3), and media-behaviors of the child with 12 items (domain 4), as presented in Table 3. The ICC was used to indicate how well the participants' ranking (i.e., from the lowest to the highest value) was on two different time points. Cohen's Kappa value was considered good indicator of interrater agreement if it was more than 0.41 [17].

## Phase 5: Internal consistency assessment

A community-based cross-sectional study was conducted among 400 primary caregivers using the final version of the DSEQ to ascertain the prevalence of DSE and its correlates in Chandigarh, Union Territory, India, from October 2017 to March 2018. Sampling frame was families having children aged 2–5 years residing in the study area. Chandigarh has a population of 1,055,450 as per census 2011, and is divided into several clusters [18]. Each cluster has a population of ranging from 4000–10000, and catered by one auxiliary nurse midwife. Auxiliary nurse midwives maintain list of families with under-five children in their catchment areas. All primary-caregivers of children aged 2–5 years living in the selected clusters were identified from the lists. All families having children aged 2–5 years, at least one TV/smartphone/digital-media gadget in their homes and consented to be a part of the study were included. Children previously diagnosed with long-term, or chronic or congenital illnesses were excluded. A multi-stage cluster random sampling technique was used to select the families. In stage one, a proportionate sampling of ten clusters representing Chandigarh's population was done. The study population was distributed into nine urban groups (90%) and one rural set (10%) [18]. In stage two, 36 families having children aged 2–5 years were randomly selected from each cluster to meet the desired sample size of 360.

The lead author interviewed the primary caregivers by face-to-face interview using the final DSEQ (modified in Phase 4) in their homes at a time convenient to them. The DSEQ was used along with standard child's sleep habits questionnaire-abbreviated [19], and standard

preschool-child behavior checklist [20] to also obtain information on sleeping habits and emotional behaviours. We have used the American Academy of Paediatric's guidelines (2016) of DSE of more than one hour per day by any electronic device to define the screen-time as excessive [9]. The detailed methodology and results of the prevalence study are given elsewhere (unpublished) [21].

**Phase 5: Data management and analysis.** The IBM SPSS Statistics for Mac, Version 25.0. Armonk, NY (2017) and Microsoft Excel version 2019 were used to estimate the internal consistency by calculating Cronbach's alpha to obtain a factor structure, which retained the maximum number of variables and explained the maximum variance. Only questions from which a response could be derived were analyzed, and leading questions were excluded from the analysis to avoid missing responses. The final questionnaire had five domains and 86-items. (S1 File). Internal consistency of 23 items from sociodemographic characteristics was not assessed as this information was obtained from the existing validated tools. Validation of 11 items from parental media literacy was also not done as these were not directly related to DSE.

We had analyzed our data in two ways based upon the nature of the dependent variable, i.e., screen-time as categorical (screen-time of less than one hour and those with more than one hour per day) and continuous (in minutes). There is evidence that screen-time varies on weekdays and weekends in the literature [22]. So, we had calculated the average DSE (by any electronic device) per day separately on a weekday and weekend.

## Ethical considerations

Postgraduate Institute of Medical Education and Research's Ethics Committee gave ethical approval (INT/IEC/2019/000711). The participant information sheet was disseminated (English and Hindi) and explained in their preferred local language. The written informed consent was obtained from the participants before initiating the study. The Director of Health Services, Chandigarh, Union Territory (VO/FW/17/1894, Dated 30/08/17) granted us permission for conducting the study in the community setting.

## Results

Phase wise results are given below.

## Phase 1: Item development

The existing literature had shown that only two studies used validated tools to measure DSE. A United-Kingdom-based study validated a tool for adults [12], whereas an Asian study validated a Chinese and English language tool for pre-schoolers [13]. A preliminary DSEQ with 250 items was developed in the English language with all possible questions per the literature review (N = 43 studies) findings. The correlates of DSE identified by the review of literature were listed at four levels; namely, a) intrapersonal level (child's demographic, behavioral, and biological factors), b) interpersonal level (interactions between the parent/caregiver and the child), c) immediate micro-environment level (home environment within which the family lives), and d) macro-environment level (socio-cultural, geographical and environmental factors) [1,14,23,24].

## Phase 2: Assessing face and content validity

The correlates identified in the review of literature were used to arrange the items into separate domains. The rearrangement of items included maintaining the flow of questions in the interviews with the caregivers after consultations with the experts. On average, the caregivers were

**Table 1. Description of the digital screen exposure questionnaire (DSEQ) items and constructs.**

| Name of the domain N = 5 | No. of items N = 86 | Description | Reasons for modifications |
|---|---|---|---|
| 1. **Sociodemographic characteristics** <br> 1.1 Demographic correlates of the family <br> 1.2 Types of gadgets available at home <br> 1.3 Type of child-care utilized with duration | 23 items <br> 15 items <br> 6 items (2 questions per gadget) <br> 2 items (3 questions per item) | Research shows associations and correlations with socio-demographic characteristics of the family so, these were incorporated. [1,25]. | This part was derived from existing tools [26] and further reworded to be accultured in the questionnaire. |
| 2. **Screen-time exposure and Home media environment** <br> 2.1 Duration, frequency of activities performed by the child and frequency of adult supervision <br> 2.2 Accessibility to these gadgets <br> 2.3 Media rules at home <br> 2.4 Duration, frequency and type of gadget used of caretakers | 27 items <br> 20 (4 subsets per 5 activities) <br> 1 item <br> 5 items (5 common media rules) <br> 1 item (4 questions per parent) | This domain helps in getting to know what all activities a child performs on a usual day. Also, caregivers can be guided to supervise children in spending more time in playful activities. This domain can also measure pre- and post-effects of an intervention. | All the activities performed by the child on a typical day were compiled in a single table to prevent the repetition of questions. |
| 3. **Level of physical activity** <br> 3.1 Weekday, weekend and total duration of outdoor activity <br> 3.2 Duration of activities performed | 10 items <br> 3 items <br> 7 items | This domain measured the level of physical activity as per the Pre-PAQ questionnaire [15] separately on a weekend or a weekday. <br> This domain can also measure pre- and post-effects of an intervention. | There were no differences in the responses of the parents separately for the weekday/ weekend activities of the children. |
| 4. **Media related behaviors** <br> 4.1 Activities performed while watching gadgets <br> 4.2 Content of media watched | 15 items <br> 3 questions (5 point Likert-type questions) <br> 12 (5 point Likert-type questions) | To measure the parenting skills and modeling appropriate behavior. <br> This domain can also measure pre- and post-effects of an intervention. | This section had questions related to only the type of content watched by children initially, the frequency was added to measure the most commonly viewed media-content. |
| 5. **Media literacy of the parents** <br> 5.1 Positive effects <br> 5.2 Negative effects | 11 items <br> 5 items <br> 6 items | This domain measures the parental perceptions on effects of screen time on the child's growth and development and also assess the literacy of the caregiver. <br> This domain can also measure pre- and post-effects of an intervention. | Open-ended questions were initially asked to aid in capturing all the responses suggested by the caregivers and to make the tool structured. So, this domain has yes/ no type answers. |

aged approximately 32 years. Mothers were interviewed in 70% of interviews. Most of the families belonged to the middle class (70%) and had a master's degree (60%). Most mothers (71.4%) and fathers (66.7%) were employed. The backgroud characterstics of the families included in this phase of the study is given in S2 Table.

A prototype DSEQ with five domains (sociodemographic; screen-time exposure and home media environment; the level of physical activity; media-related behaviors, and parental perceptions) and 86 items was developed. (Table 1).

The sociodemographic domain (personal details) had 23 items. The sociodemographic questions were obtained from an already existing validated Household Questionnaire, used in the National Family Health Survey, 2015–16, in India [26]. The second domain had 27 items related to screen-time exposure and home media environment, accessibility of gadgets, parents, and parenting approach patterns were included in the questionnaire. The third domain was associated with the level of physical activity of the child and had 10-items. This domain had questions about routine activities that the child of that age performed during weekends and weekdays. It was difficult for caregivers to keep track of all the activities and

duration. The children's media-related behaviors were captured in the fourth domain, and it had 15 items.

The initial draft had mainly open-ended questions like, '*What does the caregiver do to keep the child busy at home besides DSE*?', and '*What are the home-media rules*?' *etc*. These were later converted to multiple-choice questions as per the feedback given by the caregivers. The fifth domain had inquiries related to parental perceptions and literacy regarding DSE and had 11 items. It had questions like '*How do you react (positively and negatively) when a child behaves in a particular situation*?

For example, to measure the DSE from different digital media devices and the time spent on various activities by the child we had prepared six questions for each activity. These activities were watching smartphone, watching computer/ laptop/ tablet, reading books, being read by an adult, art and craftwork performed, and any other activities. Like for the duration of television watched by the child in a day following 6 questions were planned to be asked:

1) Does the child watch television?

2) How frequently did the child watch television in a day?

3) Duration of television watched on a working/ school day?

4) Duration of television watched on a weekend day/ holiday?

5) Does an adult supervise the child while he/she watches television?

6) How frequently does an adult supervise the child while he/she watches television?

We had later removed questions 1 and 5 as they could also be answered by questions 2 and 6, respectively. So in total we had reduced 12 items ($2*6 = 12$ items) from the questionnaire for all of the above-stated six activities. So, initially, the screen-time and home-media environment domain had 36 questions for the activities performed by the child, which were later reduced to 24 items.

Mean ratings of the content and face validation by the experts (n = 9) as per their responses on the Likert type feedback proforma are given in Table 2. The mean score for most of the feedback form questions was more than 3 (range 1 to 4). Most of the experts reported that the questionnaire would be useful to a large extent for the researcher to deal with children with excessive screen-time; caregivers to report their perceptions on DSE; covering the patterns of DSE; and for use in a heterogeneous population.

An objective measure for valid measurement of DSE was added as per the experts' suggestions. For example, questions like '*Which programs does the child usually watch on TV*?' and '*Duration of each program watched on a typical day*?' were added in the second domain. Based on all these inputs, sub-scales were developed to measure specific constructs and items within the broad domains. This was a pragmatic decision based on designing an intervention to reduce DSE among children later.

## Phase 3: Feasibility assessment

The results of the pilot study had shown that using revised DSEQ was feasible in the field settings. Children's mean age was 3.3 (±2 months) years, and nearly half (52.5%) were from upper socioeconomic status. Most (62%) participants were boys. It was observed that more than half (57%) of children on weekdays, and about 54% on weekends, had DSE of more than 60 minutes per day. The frequency of watching TV (91%) was higher than a smartphone (62%), and computers or laptops, or tablets (3%). The background characterstics of the families included in the pilot study is given in S2 Table.

Table 2. Mean ratings of the content and face validation by experts of the digital screen exposure questionnaire.

| Queries | Grading | | | | | Mean Score |
|---|---|---|---|---|---|---|
| | 0 | 1 | 2 | 3 | 4 | |
| | - | Not at all | Minimally | To certain extent | To large extent | |
| a. To what extent do you think that the questionnaire will be useful for researchers dealing with children with excessive screen-time? | - | 0 | 1 | 4 | 4 | 3.3 |
| b. To what extent do you think the questionnaire will be useful for the caregivers to report the parents perceptions/ problems related to screen-time? | - | 0 | 0 | 3 | 6 | 3.7 |
| c. To what extent do you think the questionnaire covers most patterns of usage considered to be associated with screen-time? | - | 0 | 0 | 2 | 7 | 3.8 |
| d. To what extent do you think that the questionnaire measures screen-time comprehensively? | - | 0 | 1 | 5 | 3 | 3.2 |
| e. To what extent is the language of the questionnaire appropriate and understandable? (considering the fact that screen-time is common in rural/ illiterate and urban/literate population) | - | 0 | 0 | 4 | 5 | 3.6 |
| f. How will you rate the length of the questionnaire? | Very short | Short | Adequate | Long | Very long | |
| | 0 | 0 | 2 | 7 | 0 | 2.8 |

Regarding DSEQ, although the number of items remained the same, more options regarding the specific correlates were added in the second (screen-time exposure and home media environment) and fourth domain (media-related behaviors). For example, the question on digital-media content watched by the child routinely had two options (educational content or non-educational content), that were increased to 12 options after this stage. Additional questions were type of media devices used, and child's frequency of watching the gadgets, background TV (the duration and frequency of switching it on at home). This step helped in including all the missing options per item in the questionnaire after testing it in the community.

## Phase 4: Test-retest reliability

The average age of parents was 34.2 years. On average, the families had an earning of Rs. 47250/- ($647.3 where 1 USD = Rupees 73). Half (50) of the participants had a master's degree. All (100%) fathers and 57.8% mothers were working. S2 Table.

The test-retest reliability measures, i.e., kappa values (range, 0.52–1), and ICC values (range, 0.62–0.99) for 28 items (pertaining directly to DSE), were significant ($p<0.05$) on two occasions, suggesting adequate test-retest reliability (Table 3). The inter-rater agreement for the interview coding was good (kappa 0.75, 95% CI 0.72, 0.78). The highest test-retest reliability was observed for one item, i.e., 'the duration of watching television on a typical working day' with kappa 0.89, ICC 0.97. The lowest test-retest reliability was observed for two items including; 'the average duration of outdoor play on working days/ school days with kappa 0.51' and ICC 0.69, and 'the average duration of outdoor play on holidays per day' with kappa 0.51, ICC 0.97. Hence, it can be assumed that Cohen's Kappa and ICC were within the appropriate range for measuring DSEQ and its correlates.

## Phase 5: Internal consistency assessment

The correlates of DSE were rearranged to improve the internal consistency of three domains identified in the literature search at the individual level (screen-time exposure and home media environment, media-related behaviors and physical activity). The overall internal consistency for 52 items (excluding 23 items from sociodemographic characteristics and 11 items

**Table 3. Internal consistency and reliability of questions measuring duration, frequency, patterns, and content of screen-time usage among children.**

| Domain and items | Cohen's Kappa | Intraclass correlation coefficient |
|---|---|---|
| **Domain 2: Screen time exposure and Home media environment** | | |
| 1. What is the frequency of watching television in a typical week? | 0.76$ | 0.82$ |
| 2. Duration of watching television on a typical working day? | 0.89* | 0.97$ |
| 3. Duration of watching television on a typical holiday? | 0.59$ | 0.97$ |
| 4. Does the child watch television supervision frequency by an adult? | 1$ | 0.93$ |
| 5. What is the frequency of using smartphone in a typical week? | 0.52¥ | 0.56¥ |
| 6. Duration of using smartphone on a typical working day? | 0.77$ | 0.87$ |
| 7. Duration of using smartphone on a typical holiday? | 0.6$ | 0.99$ |
| 8. Does the child us smartphone supervision frequency by an adult? | 0.62* | 0.03* |
| 9. What is the frequency of watching laptop/ computer in a typical week? | 0.84$ | 0.87$ |
| 10. Duration of watching laptop/ computer on a typical working day? | 0.73$ | 0.75$ |
| 11. Duration of watching laptop/ computer on a typical holiday? | 0.53$ | 0.87$ |
| 12. Does the child watch laptop/ computer supervision frequency by an adult? | 1$ | 0.97$ |
| 13. Do you have any rules regarding when, where, what & how to watch digital screen? | 0.84$ | 0.92$ |
| 14. Average duration of screen time per day of the caretaker | 0.6$ | 0.96$ |
| **Domain 3: Level of physical activity** | | |
| 15. Average duration of outside play per day on working/ school days | 0.51$ | 0.69$ |
| 16. Average duration on holidays of outside play per day | 0.51$ | 0.97$ |
| **Domain 4: Media behaviors of the child** | | |
| 17. The child uses digital media gadgets for completing homework assignments online | 0.64¥ | 0.66$ |
| 18. The child uses video calling applications to talk to the family/ friends | 0.69$ | 0.7$ |
| 19. The child uses digital media gadgets for learning poems, rhymes, alphabets etc. | 0.63¥ | 0.68$ |
| 20. The child uses digital media gadgets to learns math's, numbers, tables | 0.78$ | 0.79$ |
| 21. The child uses digital media gadgets to recognize shapes/ sounds/ colors | 0.83$ | 0.84$ |
| 22. The child uses digital media gadgets to learns various sciences online | 0.62¥ | 0.62$ |
| 23. The child uses digital media gadgets to learns to draw/ write | 0.62¥ | 0.62$ |
| 24. The child plays video-games on digital media gadgets | 0.65$ | 0.66$ |
| 25. The child uses digital media gadgets to watches stories | 0.83$ | 0.84$ |
| 26. The child uses digital media gadgets to watch adult programs (soap opera, news, sports, movies etc.) on media screens online | 0.67$ | 0.68$ |
| 27. The child uses digital media gadgets to learns letters, words, vocabulary, language online | 0.63¥ | 0.63$ |
| 28. Digital media gadgets to watch random things for enjoyment (music, advertisements, click photos etc.) | 0.7$ | 0.7$ |

*level of significance = <0.05;

¥ the level of significance = <0.01;

†level of significance = <0.001;

$ level of significance = <0.0001.

from parental media literacy) was adequate at a 95% confidence interval with Cronbach's alpha of 0.71. The correlates identified in these domains were matched for inclusion or exclusion in order to maintain the context and flow of questions. A separate domain-wise analysis for 27 items of screen-time exposure and home media environment showed a Cronbach's alpha of 0.82. The media-related behaviors domain with 15 items had a Cronbach's alpha value of 0.74. The Cronbach's alpha was 0.73 for ten items of physical activity.

The backgroud characterstics of the families included in the prevalence study is given in S2 Table. Most of the families were belonging to the above middle socioeconomic class (55.8%). The majority of the mothers were homemakers (86%), and fathers worked as unskilled/semi-skilled/skilled workers (68%). Nearly half (52.6%) of the mothers and fathers (55.7%) were high school graduates and/or beyond. On an average, 59.5% of children (mean age 3.5±0.9 years) had excessive screen-time. It was higher on weekdays (58.5%) as compared to the weekends (56.8%). The results of the sleep and emotional behaviors have not been presented in this manuscript but published elsewhere [21]. The unpublished results of that study have shown that standard child's sleep habits questionnaire-abbreviated [19] and standard preschool-child behavior checklist [20] can also be used along with DSEQ to estimate DSE's associations with sleeping habits and emotional behaviours.

## Discussion

This paper reports the five inter-related studies that focused on developing and testing the psychometric properties of a DSEQ and its correlates among children two to five years of age. This study's results supported the content validity, test-retest reliability, and internal consistency of the DSEQ. Th validated DSEQ tool will be useful in measuring DSE, and its correlates among young children in India, who have the most significant growing telecom network with approximately 875 million users [27].

Caregivers play a pivotal role managing their child's recreational time [13]; hence, the DSEQ was designed to measure DSE from all types of gadgets and its correlates as reported by the primary caregiver. The tool also captured parental literacy and perceptions to understand how parents influence a child's motivation or demotivation to watch digital-media [3,13]. Due to the emphasis on the placement of digital-media gadgets at home, we dedicated 15 items (media-related behaviors) to this aspect [13]. Literature states that DSE in children should be the cumulative time of all the gadgets viewed in a day [3]; hence, a particular time for all devices used was added to the total DSE. Children preferred certain physical activities more than the others; therefore, separate weekday and weekend questions seemed irrelevant [15]. A study [28] reported that the length and complexity of a questionnaire could have adverse effects on the results; thus, questions were rechecked for appropriateness after each phase. The main prevalence study conducted in the last phase also included two separate behavior scales on sleep and emotional disorder [21]. The results of phase five indicate that DSEQ can also be used with the standard child's sleep habits questionnaire-abbreviated [19] and standard preschool-child behavior checklist [20] to measure the association of DSE with sleep habits and emotional behaviours.

Indian children live with their natural caregivers [29] so, this study was primarily planned in the home-environment in a community setting. Each family owned at least one gadget [21], which was seen in 98% of Hong-Kong families [13]. There was a difference observed in weekday and weekend DSE in existing literature [22]; so, we measured DSE separately for weekdays and weekends. The results of the pilot study were consistent with the main prevalence study [21]. This suggests the replicability of the questionnaire. Questions on the home-media environment, parental perceptions, and literacy-related to DSE were explicitly added to be

modified later in the intervention study [13]. It can be used to identify the modifiable (psycho-social and behavioral) determinants of DSE in young children.

The DSEQ measured the desired outcomes domain-wise. The scores were not adjusted for each family as personal characteristics did not impact these scores, as seen in previous studies [30]. However, parenting skills, sociodemographic background, and the type of media regulations might influence caregiving among families [1]. The questionnaire had both English and Hindi languages written side by side. These bilingual questionnaires are commonly used in nationwide surveys in India [26]. The inter-rater agreement for the interview coding (kappa 0.75, 95% CI 0.72, 0.78), test-retest reliability statistics (kappa values, and ICCs) of individual items, and internal consistency determined by Cronbach's alpha of the DSEQ were substantial [30].

To the best of our knowledge, this is the first study to describe the reliability and consistency of the DSEQ to be used among children aged 2 to 5 years. As per Hawkins et al (2019) a valid questionnaire/ tool can be developed by evidence based on the a) content, b) response processes, c) internal structure, d) external variables, and e) validity and the consequences of testing [31]. All these parameters were met while developing the DSEQ in this study, which is a significant strength. Multiphasic development of the questionnaire helped us to modify the tool at each stage before being used in the next phase and at community level. An extensive literature search led to designing and testing a robust tool for measuring DSE in Indian children. The methodology also incorporated the qualitative reviews of the experienced experts and the caregivers for refining the questionnaire [3,30]. McHugh et al. (2012) suggested that the inter-rater reliability can be assumed to be appropriate if Cohen's Kappa is above 0.41, and minimum of 30 observations are made [16]. Cohen's Kappa value is also the most acceptable value for healthcare settings considering the possibility of guessing. In our study, we had 30 comparisons in the test retest phase, and minimum Cohen's kappa was found to be above 0.51, which indicated that it is reliable. DSEQ developed in this study comprehensively measures DSE and its relevant correlates, while the existing tools only measure either the parent and child media-addiction or frequency of DSE or emotional problems [13] or duration of DSE [11,12] and background screen use [11]. Lastly, the results of this study can be generalized to an area with both urban and rural settings, as the last phase of the study was conducted in representative sample size (N = 400) from Chandigarh a North Indian Union Territory with both the settings [18].

There is a certain degree of methodological limitations that might restrict the generalizability of DSEQ. Given the children's limited cognitive development, all results were reported by the child's caregivers/parents. The variability of caregivers' parenting skills and perception was unusual and could be an artifact in bringing behavior change in children. The Indian literature in this regard is scarce. Although there was a preponderance of mothers as primary caregivers of the child in most studies, others stated that some families used child-care services [1]. The sample size for test-retest reliability (n = 30) was small but comparable to those of another Indian study [30]. The American Academy of Pediatrics guidelines, 2016, were used to develop the tool as none existed for low- and middle-income countries including India. However, the World Health Organization has also given the similar DSE guidelines which might be applicable in low and middle income countries [9]. Lastly, most of the constructs that we had employed were quantitative in nature, directly observable and gave direct outcomes rather than latent ones. Therefore, Structural Equation Model was not employed as per the above stated conditions [31]. The method and procedure used by us for the validation of the tool in the present study is widely accepted and used by the researchers in their respective studies for the purpose of development and validation of the study tools [3,30].

## Conclusions

To conclude we developed a reliable and an internally consistent DSEQ through a rigorous methodology involving five stages including review of literature, face and content validity of the draft Hindi and English questionnaire, pilot testing, assessing test-retest reliability and internal consistency. This questionnaire can be widely used in middle and low income countries by the researchers to generate evidence on the burden of DSE among young children. Researchers can also use three domains namely, home-media environment, media behaviors of the child, and physical activity individually, as their internal consistency was measured separately. The questionnaire measured the DSE of children and its correlates consistently in the pilot study and main prevalence study. This questionnaire might be used in both epidemiological and intervention studies to estimate the prevalence of DSE from all types of media devices. The public health implications of this study are that the robust estimates of digital screen exposure using the validated DSEQ among young children might be generated that might influence the policy-makers and Pediatric associations to formulate guidelines on permissible limits of DSE among children in low- and middle-income countries. Future studies should measure DSE and its correlates with DSEQ in varied populations so that it can be further tested.

## Supporting information

**S1 Table. Likert-type feedback proforma for assessing the face and content validity of the DSEQ.**
(DOCX)

**S2 Table. Socio-demographic characteristics of the participants for various phases.**
(DOCX)

**S1 File. Digital screen exposure questionnaire (DSEQ).**
(DOCX)

## Author Contributions

**Conceptualization:** Nimran Kaur, Madhu Gupta, Prahbhjot Malhi.

**Data curation:** Nimran Kaur, Madhu Gupta, Tanvi Kiran.

**Formal analysis:** Nimran Kaur.

**Funding acquisition:** Nimran Kaur, Madhu Gupta.

**Investigation:** Nimran Kaur.

**Methodology:** Nimran Kaur, Madhu Gupta, Tanvi Kiran, Prahbhjot Malhi, Sandeep Grover.

**Project administration:** Nimran Kaur.

**Resources:** Nimran Kaur.

**Software:** Nimran Kaur.

**Supervision:** Madhu Gupta, Tanvi Kiran, Prahbhjot Malhi, Sandeep Grover.

**Validation:** Nimran Kaur, Tanvi Kiran.

**Visualization:** Madhu Gupta, Tanvi Kiran.

**Writing – original draft:** Nimran Kaur.

**Writing – review & editing:** Madhu Gupta, Sandeep Grover.

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
