## [Editor Report · Decision Letter 0]

6 Jan 2021

PONE-D-20-22585

Development and Evaluation of the Digital-Screen Exposure Questionnaire (DSEQ) for Young Children

PLOS ONE

Dear Dr. Gupta,

Thank you for submitting your manuscript to PLOS ONE. After careful consideration, we feel that it has merit but does not fully meet PLOS ONE’s publication criteria as it currently stands. Therefore, we invite you to submit a revised version of the manuscript that addresses the points raised during the review process.

We look forward to receiving your revised manuscript.

Kind regards,

Adewale L. Oyeyemi, Ph.D

Academic Editor

PLOS ONE

Additional Editor Comments:

1. Abstract, Methods: Include the “n” to indicate the number of the randomly selected family in the phase 3 of the questionnaire development.

2. Abstract, Methods: The last sentence “It had strong internal consistency (Cronbach’s alpha 0.73-0.82) regarding digital screen exposure-related items” is not clear. Is it only the screen exposure and home media environment domain that had strong internal consistency or all the domains?

3. Abstract, Conclusion: For clarity, the author can revise the conclusion as: The developed DSEQ has good face and content validity and acceptable evidence of internally consistency and test-retest reliability. The DSEQ can be used digital screen exposure and its correlates among children aged 2 to 5 years.

4. Introduction, Paragraph 1, first sentence. Please clarify what you imply by the words “have been stimulated” in the first sentence.

5. Introduction, Last Paragraph: A new last paragraph should be included to clearly elucidate on the justification and rationale of the study. For example, what is the benefits and importance of developing the DSEQ for research and practice in India. Also, the objectives of the study need to be clearly stated.

6. Methods, Phase 1, last sentence: Please clarify that these are correlates of Digital Screen Exposure identified in the literature and not correlates of the DSEQ developed and tested in your study.

7. Methods, Phase 2: In the abstract, the Phase 2 stage started with the experts and followed by the primary care givers but in the methods, phase 2 started with the primary care givers then followed by the experts. Also, how many of the caregivers were interviewed in each of the four rounds of face-to-face interview with the primary caregivers. At present, it is not clear if all the 30 primary caregivers were involved at the same time in each of the four rounds of interview (FGD?) or few were selected per each round of the interview? There is also a conflict between the number of experts that participated in stage 2 as stated in the abstract and figure 1 (n=9) on one hand and in the methods (n=3 from psychology, psychiatry & public health + n=25 experts through mail or personal meeting) on the other hand. Moreover, what does this statement means: Nine experts from pediatrics, public health, community medicine, psychology, psychiatry, and pediatric neurology reverted.

8. Methods, Phase 3: It is not clear what was done in the feasibility study. Was the prototype questionnaire administered to the children? If so, what was the mode of administration? Self-administered or interviewer administered? Did the children complete the questionnaire directly or by proxy (i.e., completed by the parents)?

9. Methods, Phase 4: Provide more information on the description of the random selection methods used to select the participants in the test-retest reliability study. What was the sampling frame of the population? Also, provide information on how the interrater agreement for the interview was determined in the data management and analysis.

10. Methods, Phase 5: Please include a brief description of the methodology of the Internal Consistency Assessment phase of the study.

11. Results, Phase 2, Last Paragraph: Please include examples of the subscales that were developed to measure specific constructs and items within the broad domain.

12. Results, Phase 4: Provide the information for the item with lowest and highest Kappa values and ICCs. This way, it is easy for the reader to have a sense of the items with the lowest and highest reliability coefficients without referring to the tables.

13. Discussion, First Paragraph, First Sentence: It was alluded here that the study focused on developing and testing the psychometric properties of a DSEQ and its correlates among children. However, there is no where in the results where the findings on correlates of the DSEQ was reported. The author shall rather include the correlates results in the manuscript or eliminate this as the focus of their study throughout the manuscript.

14. Discussion, Second Paragraph, Last Sentence: How can it be asserted that DSEQ can be used with other psychological and social scale when this was not tested in your study?

15. Discussion, Third paragraph, Third Sentence: It was stated that results of the pilot study was consistent with the main prevalence study that has been unpublished. It is important to include the results of the main study here for easy comparison.

16. Discussion, Fourth Paragraph: The authors should elucidate further on the potential reasons for their findings of substantial and acceptable psychometric coefficients for most of the items on the DSEQ in the study population.

17. The manuscript ended abruptly without a conclusion. Please include a brief section on Conclusion to summarize the implication of the main findings (just as in the abstract) and the direction for future research in this context.

Journal Requirements:

2. In your Methods section, please provide additional information about the demographic details of your participants. Please ensure you have provided sufficient details to replicate the analyses such as: a)  a description of any inclusion/exclusion criteria that were applied to participant inclusion in the analysis, b) a table of relevant demographic details, c) a statement as to whether your sample can be considered representative of a larger population. In addition, please state what type of consent was obtained from the parents of children included in this study (for instance, written or verbal, and if verbal, how it was documented and witnessed).

---

## [Author Response · Author response to Decision Letter 0]

20 Feb 2021

Responses to reviewers comments

Comments Responses Page and line number

1. Abstract, Methods: Include the “n” to indicate the number of the randomly selected family in the phase 3 of the questionnaire development. The ‘n=40’ has been now added to indicate the number of the randomly selected families in the phase 3 of the questionnaire development as suggested by the reviewers. Page 3, line 56

2. Abstract, Methods: The last sentence “It had strong internal consistency (Cronbach’s alpha 0.73-0.82) regarding digital screen exposure-related items” is not clear. Is it only the screen exposure and home media environment domain that had strong internal consistency or all the domains? This line pertains to abstract’s results. 

It is clarified that the DSEQ had five domains, including sociodemographic, screen-time exposure and home media environment, level of physical activity, media-related behaviors, and parental perceptions. This line is revised as follows:

‘A strong internal consistency was observed for three domains including, screen-time exposure and home media environment (Cronbach's alpha of 0.82), physical activity (Cronbach's alpha 0.73) and media-related behaviors (Cronbach's alpha of 0.74).’ Page 4, line 68-71

3. Abstract, Conclusion: For clarity, the author can revise the conclusion as: The developed DSEQ has good face and content validity and acceptable evidence of internally consistency and test-retest reliability. The DSEQ can be used digital screen exposure and its correlates among children aged 2 to 5 years. We have revised the conclusion in the abstract as suggested. Page 4, line 73-76

4. Introduction, Paragraph 1, first sentence. Please clarify what you imply by the words “have been stimulated” in the first sentence. We have rephrased this sentence as follows:

‘Caregivers regulate the digital ecosystem within which their children grow up [1]. A child learns new skills by impersonating their parents or family members regarding media-use [2].’

Page 5, line 81-82

5. Introduction, Last Paragraph: A new last paragraph should be included to clearly elucidate on the justification and rationale of the study. For example, what is the benefits and importance of developing the DSEQ for research and practice in India. Also, the objectives of the study need to be clearly stated. A new paragraph at the end of the introduction that includes the justification, rationale and objectives of the study has been incorporated.

‘A review by Kaur et al. (2019) suggested that there is a need to assess the DSE’s prevalence among under-five children in the developing countries including India to understand how they use digital-media devices [1]. However, it was reported that only limited validated tools existed to measure DSE [11–13]. Among these tools one was in Chinese and the other two in English language [1]. The existing tools were developed for older age-groups (3-6 years [13], 5-15 years [12], and above 18 years [11]). There was no tool that comprehensively measured DSE and its correlates among toddlers including intrapersonal, interpersonal, immediate media micro-environment and macro-environment associates. Measuring a young child’s DSE and its correlates with a validated tool is important as DSE might be associated with screen-based sedentary behaviours affecting health outcomes like emotional behaviours [6,10], sleep problems [14] in the early childhood or development of non-communicable diseases during the adolescense [9] and adulthood [4]. As the media-exposure access and usage patterns are different among different populations so the country-specific data needs to be generated.

This study’s objectives were to develop a comprehensive DSE questionnaire (DSEQ) for children aged 2-5 years to assess the cumulative DSE (including all types of digital-media devices used by the child routinely) and its correlates (including parental digital-media behaviors, parental digital-media literacy, home-media environment, media-content watched by the child, physical activity of the child, activities performed by the child, etc.). This study’s findings might help the researchers to assess the burden of DSE, especially in the developing countries, and impact of interventions to reduce the DSE using a validated tool, and help in formulating the age-specific policies to regulate the DSE among preschoolers.’ Page 6, line 107-128

6. Methods, results Phase 1, last sentence: Please clarify that these are correlates of Digital Screen Exposure identified in the literature and not correlates of the DSEQ developed and tested in your study. This line pertains to the manuscript’s results. 

We have clarified that the correlates of DSE were identified in the literature.

‘The correlates of DSE identified by the review of literature were listed at four levels, namely, a) intrapersonal level (child’s demographic, behavioral, and biological factors), b) interpersonal level (interactions between the parent/caregiver and the child), c) immediate micro-environment level (home environment within which the family lives), and d) macro-environment level (socio-cultural, geographical and environmental factors) [1,22–24].’

Page 14, line 311-316

7. Methods, Phase 2: In the abstract, the Phase 2 stage started with the experts and followed by the primary care givers but in the methods, phase 2 started with the primary care givers then followed by the experts. 

Also, how many of the caregivers were interviewed in each of the four rounds of face-to-face interview with the primary caregivers. 

At present, it is not clear if all the 30 primary caregivers were involved at the same time in each of the four rounds of interview (FGD?) or few were selected per each round of the interview? 

There is also a conflict between the number of experts that participated in stage 2 as stated in the abstract and figure 1 (n=9) on one hand and in the methods (n=3 from psychology, psychiatry & public health + n=25 experts through mail or personal meeting) on the other hand. Moreover, what does this statement means: Nine experts from pediatrics, public health, community medicine, psychology, psychiatry, and pediatric neurology reverted. It is clarified that Phase 2 stage started with the primary caregivers and followed by the experts. It is now revised in the abstract.

The paragraph has been elaborated for better understanding as follows: 

6-8 caregivers were interviewed per round in four rounds. So, we interviewed 30 primary caregivers in total by face to face interview.

Each primary caregiver was interviewed separately by face-to-face interview. It was not a FGD.

3 experts (who were also the authors) from the field of public health, community medicine, psychology, psychiatry, were consulted in the initial phase of face to face interview of the primary caregivers, regarding addition, deletion rearrangement of the questions and modifying the DSEQ accordingly for acculturation.

Later 25 different experts from the fields of pediatrics, public health, community medicine, psychology, psychiatry, and pediatric neurology, were contacted. They had at least five years of experience and a doctor’s degree (Doctor of Medicine/Ph.D.). They were approached through the mail or personal meetings to assess the content (structural construct, utility/ futility of questions, arrangement/ flow, sections/ sub-sections, and domain-wise distribution) and face validity (overall sequence/ language and appropriateness). Of these, 9 experts reverted with their feedback.

This is now clarified in the Methods section under Phase 2: Assessing face and content validity

‘Initially, four rounds of the face-to-face interviews with 30 primary-caregivers (6-8 per round) of children aged 2-5 years, attending the pediatric outpatient department of a tertiary-care-hospital in Chandigarh were conducted in a separate room for acculturation of DSEQ. Participants who were at least middle school pass outs, and were chosen to contribute to modify the questionnaire. The lead author interviewed the primary-caregivers individually using the preliminary DSEQ (English version) in July 2017. In each interview, the parents were asked about their understanding of digital-media usage in children and any missing areas or concerns in the questionnaire. After each round of face-to-face interviews, the draft DSEQ was modified in consultation with three experts (authors themselves), one each from the specialty of psychology, psychiatry, and public health. These experts by rewording, addition, deletion, and rearrangement of the questions modified the DSEQ for acculturation. New and modified questions were used to interview a new set of primary caregivers in each round, until all possible questions/responses were included. The rearrangement, addition or removal of an item was checked by two experts independently, and finalized by a third one in case of any disagreement. The questionnaire was sequentially translated into the Hindi language by bilingual health professionals (n=3) using standard translation-back translation methodology [15]. 

Later, the bilingual DSEQ was given to 25 different experts from the fields of pediatrics, public health, community medicine, psychology, psychiatry, and pediatric neurology. They had at least five years of experience and a doctor’s degree (Doctor of Medicine/Ph.D.). These experts were approached through the mail or personal meetings to assess the content (structural construct, utility/ futility of questions, arrangement/ flow, sections/ sub-sections, and domain-wise distribution) and face validity (overall sequence/ language and appropriateness). The experts were given the conceptual definitions of the underlying constructs, the study’s objectives, and a Likert-type feedback proforma. (S2 Table). Each question’s score in the feedback proforma ranged from 1 to 4 where, 1 = ‘not at all’, 2 = ‘minimally’, 3 = ‘to a certain extent’, and 4 = ‘to a large extent’. The score on the length of the questionnaire ranged from 0 to 4, where 0 = ‘very short’, 1= ‘short’, 2 = ‘adequate’, 3 = ‘long’, and 4 = ‘very long’. Qualitative comments on the questions were also asked. Of the 25 experts contacted, nine experts reverted. The questionnaire was further modified, as per the comments of the experts.’ Page 7-9, line 149-179

8. Methods, Phase 3: It is not clear what was done in the feasibility study. 

Was the prototype questionnaire administered to the children? 

If so, what was the mode of administration? 

Self-administered or interviewer administered? 

Did the children complete the questionnaire directly or by proxy (i.e., completed by the parents)? It is clarified that in the feasibility study, a pilot study was conducted in the community settings to assess how feasible was to use the questionnaire (DSEQ) in the home environment of the families.

No, it was administered to their primary caregivers. The information regarding the children was proxy-reported by their primary caregiver’s.

Mode of administration was the face to face interview of the primary caregivers by visiting their homes

It was interviewer administered

The information regarding the children was proxy-reported by their primary caregiver’s.

The related text in the manuscript is now revised as follows:

For feasibility assessment, ‘a pilot study was conducted to assess how feasible was to use the prototype DSEQ in the home environment of the families in the community. This study was conducted in the urban settings in Sahib Ajit Singh Nagar, district Mohali, Punjab, India, in August 2017. All families having children aged 2-5 years, at least one TV/smartphone/digital-media gadget in their homes and consented to be a part of the study were included. Children previously diagnosed with long-term, or chronic or congenital illnesses were excluded. A multi-stage simple random sampling was done to select 40 random families from the study area. In the first stage, a catchment population area of one auxiliary-nurse-midwife in Phase I, Sahib Ajit Singh Nagar was randomly selected. In the second stage, the families with children between 2-5 years were randomly selected from the catchment area by computer-generated random numbers. The selected families were then contacted by the lead author by visiting their homes to conduct the face to face interviews with the primary caregivers of the children using the prototype DSEQ (developed in phase 2), at a time convenient to them. The information regarding the children was proxy-reported by their primary caregivers.’ Page 9

Line 189-202

9. Methods, Phase 4: Provide more information on the description of the random selection methods used to select the participants in the test-retest reliability study. 

What was the sampling frame of the population? 

Also, provide information on how the interrater agreement for the interview was determined in the data management and analysis. Phase 4 of the study was also conducted in the community, but in different location than the pilot study, i.e., in Phase III, Sahib Ajit Singh Nagar, district Mohali, Punjab, India

Sampling frame was families with children aged 2-5 years residing in the Phase III, Sahib Ajit Singh Nagar, district Mohali, Punjab, India. These families were selected by multistage random sampling. This is mentioned in the text as follows:

‘The test-retest reliability was done among randomly selected primary caregivers of children aged 2-5 years in September 2017, to assess the psychometric properties of the DSEQ. This stage was conducted in urban settings in Phase III, Sahib Ajit Singh Nagar, district Mohali, Punjab, India. Sampling frame was families with children aged 2-5 years residing in the study area. All the families having children aged 2-5 years, at least one TV/smartphone/digital-media gadget in their homes and consented to be a part of the study were included. Children previously diagnosed with long-term, or chronic or congenital illnesses were excluded. A multi-stage simple random sampling technique was used to select 30 families from the study area. In the first stage, a catchment population area of one auxiliary-nurse-midwife in Phase III, Sahib Ajit Singh Nagar, Punjab was randomly selected. In the second stage, the families with children between 2-5 years were randomly selected from the catchment area by computer-generated random numbers. The chosen families were contacted at their homes. The lead author interviewed the primary caregivers by face to face interview using the prototype DSEQ (modified in Phase 3), at time convenient to them. The information regarding the children was proxy-reported by their primary caregivers. 

The data’s repeatability was observed on two occasions (time one and time two), ten days apart. At time one, the lead author interviewed the willing primary-caregivers in their homes. At time two, ten days later, these participants were again approached by the lead author to assess the differences in their responses.’

In data management and analysis it now mentioned that:

‘Microsoft Excel version 2019 was used for data entry. The IBM SPSS Statistics for Mac, Version 25.0. Armonk, NY (2017) was used for measuring Cohen’s Kappa and Intraclass Correlation Coefficient (ICC) values. The results were estimated at a 95% confidence interval. We had measured Cohen’s Kappa values and Intraclass Correlation Coefficient (ICC) for three domains of DSEQ including, screen-time exposure and home media environment with 16 items (domain 2), level of physical activity with 2 items (domain 3), and media-behaviors of the child with 12 items (domain 4), as presented in Table 3. The ICC was used to indicate how well the participants’ ranking (i.e., from the lowest to the highest value) was on two different time points. Cohen’s Kappa value was considered good indicator of interrater agreement if it was more than 0.41 [16] and our study had items with a value above 0.51. Page 10-11

Line 216-235

Line 239-247

10. Methods, Phase 5: Please include a brief description of the methodology of the Internal Consistency Assessment phase of the study. Brief methodology of the Phase 5 is now included in the manuscript as follows: 

‘A community-based cross-sectional study was conducted among 400 primary caregivers using the final version of the DSEQ to ascertain the prevalence of DSE and its correlates in Chandigarh, Union Territory, India, from October 2017 to March 2018. Sampling frame was families having children aged 2-5 years residing in the study area. Chandigarh has a population of 1,055,450 as per census 2011, and is divided into several clusters [18]. Each cluster has a population of ranging from 4000-10000, and catered by one auxiliary nurse midwife. Auxiliary nurse midwives maintain list of families with under-five children in their catchment areas. All primary-caregivers of children aged 2-5 years living in the selected clusters were identified from the lists. All families having children aged 2-5 years, at least one TV/smartphone/digital-media gadget in their homes and consented to be a part of the study were included. Children previously diagnosed with long-term, or chronic or congenital illnesses were excluded. A multi-stage cluster random sampling technique was used to select the families. In stage one, a proportionate sampling of ten clusters representing Chandigarh's population was done. The study population was distributed into nine urban groups (90%) and one rural set (10%) [17]. In stage two, 36 families having children aged 2-5 years were randomly selected from each cluster to meet the desired sample size of 360. The lead author interviewed the primary caregivers by face to face interview using the final DSEQ (modified in Phase 4) in their homes at a time convenient to them. The DSEQ was used along with standard child’s sleep habits questionnaire-abbreviated [18] and standard preschool-child behavior checklist [19] to also obtain information on sleeping habits and emotional behaviours. We have used the American Academy of Paediatric’s guidelines (2016) of DSE of more than one hour per day by any electronic device to define the screen-time as excessive [9]. The detailed methodology and results of the prevalence study are given elsewhere (unpublished) [20].’

Page 11-12

11. Results, Phase 2, Last Paragraph: Please include examples of the subscales that were developed to measure specific constructs and items within the broad domain. An example of modified items have been added in the Phase 2 as suggested. 

‘For example, to measure the screen time from different sources of digital media devices and time spent on various activities by the child we had prepared six questions for each activity. These activities were watching smartphone, watching computer/ laptop/ tablet, reading books, being read by an adult, art and craftwork performed, and any other activities. Like for the duration of television watched by the child in a day following 6 questions were planned to be asked: 

1) Does the child watch television? 

2) How frequently did the child watch television in a day? 

3) Duration of television watched on a working/ school day?

4) Duration of television watched on a weekend day/ holiday? 

5) Does an adult supervise the child while he/she watches television? 

6) How frequently does an adult supervise the child while he/she watches television?

We had later removed questions 1 and 5 as they could also be answered by questions 2 and 6, respectively. So in total we had reduced 12 items (2*6=12 items) from the questionnaire for all of the above-stated six activities. So, initially, the screen-time and home-media environment domain had 36 questions for the activities performed by the child, which were later reduced to 24 items.’ Page 16, line 353-368

12. Results, Phase 4: Provide the information for the item with lowest and highest Kappa values and ICCs. This way, it is easy for the reader to have a sense of the items with the lowest and highest reliability coefficients without referring to the tables. Thank you for your suggestions. We have now included this information, in Phase 4

‘The highest test-retest reliability was observed for the item ‘the duration of watching television on a typical working day’ with kappa value of 0.89 and ICC 0.97. The lowest test-retest reliability was observed for two items including, ‘the average duration of outside play on working days/ school days’ with kappa value of 0.51 and ICC 0.69 and ‘the average duration of outside play on holidays per day’ with kappa value of 0.51 and ICC 0.97.’ 

Page 18

13. Discussion, First Paragraph, First Sentence: It was alluded here that the study focused on developing and testing the psychometric properties of a DSEQ and its correlates among children. However, there is nowhere in the results where the findings on correlates of the DSEQ was reported. The author shall rather include the correlates results in the manuscript or eliminate this as the focus of their study throughout the manuscript. The results of the correlates are included in each phase as follows:

Phase 1

‘The correlates of DSE identified by the review of literature were listed at four levels, namely, a) intrapersonal level (child’s demographic, behavioral, and biological factors), b) interpersonal level (interactions between the parent/caregiver and the child), c) immediate micro-environment level (home environment within which the family lives), and d) macro-environment level (socio-cultural, geographical and environmental factors) [1,22–24].’ 

Phase 2

‘The correlates identified in the review of literature were used to arrange the items into separate domains. The rearrangement of items included maintaining the flow of questions in the interviews with the caregivers after consultations with the experts.’

Phase 3

‘Regarding DSEQ, although the number of items remained the same, more options per item regarding the correlates were added in the second (screen-time exposure and home media environment) and fourth domain (media-related behaviors). For example, the question on digital-media content watched by the child routinely had two options (educational content or non-educational content), that were increased to 12 options after this stage. Additional questions were type of media devices used, and child’s frequency of watching the gadgets, background TV (the duration and frequency of switching it on at home). This step helped in including all the missing options per item in the questionnaire after testing it in the community.’

Phase 5

‘The correlates of DSE were rearranged to improve the internal consistency of three domains, identified in the literature, at the individual level (screen-time exposure and home media environment, media-related behaviors and physical activity). The overall internal consistency for 52 items (excluding 23 items from sociodemographic characteristics and 11 items from parental media literacy) was adequate at a 95% confidence interval with Cronbach's alpha of 0.71. The correlates identified in these domains were matched for inclusion or exclusion in order to maintain the context and flow of questions. A separate domain-wise analysis for 27 items of screen-time exposure and home media environment showed a Cronbach's alpha of 0.82. The media-related behaviors domain with 15 items had a Cronbach's alpha value of 0.74. The Cronbach's alpha was 0.73 for ten items of physical activity.’

The information on the correlates of DSE has been added in the discussion of the manuscript as suggested.

This paper reports five inter-related studies that focused on the developing and testing the psychometric properties of a DSEQ and its correlates among children two to five years of age. Parental intervention, media-literacy, and behaviors were the prime correlates that may directly or indirectly modify the home-media environment, activities performed by the child, media-behaviors, and content watched by their children. 

Page 14, line 311-316

Page 17, line 320-322

Page 18, line 398-405

Page 19, line 430-439

14. Discussion, Second Paragraph, Last Sentence: How can it be asserted that DSEQ can be used with other psychological and social scale when this was not tested in your study? In the results of phase 5, we had used DSEQ with two other questionnaires ‘the standard child’s sleep habits questionnaire-abbreviated [19] and standard preschool-child behavior checklist [20].’ The unpublished results of Phase 5 study have shown that the other psychological and social scale can be used simultaneously. 

That is why we have incorporated this in the discussion section. Page 19, line 447-451

Page 21

15. Discussion, Third paragraph, Third Sentence: It was stated that results of the pilot study was consistent with the main prevalence study that has been unpublished. It is important to include the results of the main study here for easy comparison. The brief results of the prevalence study have been incorporated in the results section for easy comparison.

Results of the pilot study:

‘It was observed that 57% of children on weekdays, and about 54% on weekends, had DSE of more than 60 minutes per day. The frequency of watching TV (91%) was higher than a smartphone (62%), and computers or laptops, or tablets (3%). The backgroud characterstics of the families included in the prevalence study is given in S5 Table.’

Results of the main prevalence study

‘On average, 59.5% of children had excessive screen-time. It was higher on weekdays (58.5%) as compared to the weekends (56.8%). The backgroud characterstics of the families included in the prevalence study is given in S7 Table.’ 

Page 17, line 388-394

Page 19, line 441-451

16. Discussion, Fourth Paragraph: The authors should elucidate further on the potential reasons for their findings of substantial and acceptable psychometric coefficients for most of the items on the DSEQ in the study population. We have now elucidated further the reasons for substantial and acceptable psychometric coefficients for most of the items on the DSEQ in the study population. 

‘To the best of our knowledge, this is the first study to describe the reliability and consistency of the DSEQ to be used among children aged 2 to 5 years. As per Hawkins et al (2019) a valid questionnaire/ tool can be developed by evidence based on the a) content, b) response processes, c) internal structure, d) external variables, and e) validity and the consequences of testing [30]. All these parameters were met while developing the DSEQ in this study, which is a significant strength. Multiphasic development of the questionnaire helped us to modify the tool at each stage before being used in the next phase and at community level. An extensive literature search led to designing and testing a robust tool for measuring DSE in Indian children. The methodology also incorporated the qualitative reviews of the experienced experts and the caregivers for refining the questionnaire [3,29]. McHugh et al. (2012) suggested that the inter-rater reliability can be assumed to be appropriate if Cohen’s Kappa is above 0.41, and minimum of 30 observations are made [16]. Cohen’s Kappa value is also the most acceptable value for healthcare settings considering the possibility of guessing. In our study, we had 30 comparisons in the test retest phase, and minimum Cohen’s kappa was found to be above 0.51, which indicated that it is reliable. DSEQ developed in this study comprehensively measures DSE and its relevant correlates, while the existing tools only measure either the parent and child media-addiction or frequency of DSE or emotional problems [13] or duration of DSE [11,12] and background screen use [11]. Lastly, the results of this study can be generalized to an area with both urban and rural settings, as the last phase of the study was conducted in representative sample size (n=400) form Chandigarh a North Indian Union Territory with both the settings.’ Page 22, line 504-524

17. The manuscript ended abruptly without a conclusion. Please include a brief section on Conclusion to summarize the implication of the main findings (just as in the abstract) and the direction for future research in this context. We agree, so a conclusion has been added in the end.

‘To conclude we developed a reliable and an internally consistent DSEQ through a rigorous methodology involving five stages including review of literature, face and content validity of the draft Hindi and English questionnaire, pilot testing, assessing test-retest reliability and internal consistency. This questionnaire can be widely used in middle and low income countries by the researchers to generate evidence on the burden of DSE among young children. Researchers can also use three domains namely, home-media environment, media behaviors of the child, and physical activity individually, as their internal consistency was measured separately. The questionnaire measured the DSE of children and its correlates consistently in the pilot study and main prevalence study. This questionnaire might be used in both epidemiological and intervention studies to estimate the prevalence of DSE from all types of media devices. The public health implications of this study are that the robust estimates of digital screen exposure using the validated DSEQ among young children might be generated that might influence the policy-makers and pediatric associations to formulate guidelines on permissible limits of DSE among children in low- and middle-income countries. Future studies should measure DSE and its correlates with DSEQ in varied populations so that it can be further tested.’ Page 23, line 539-554

Editors comments Responses Page and line number

https://journals.plos.org/plosone/s/file?id=ba62/PLOSOne_formatting_sample_title_authors_affiliations.pdf The suggested formatting have been done in the manuscript. 

2. In your Methods section, please provide additional information about the demographic details of your participants. Following tables on demographics are added:

S4 Table: Socio-demographic characteristics of the participants for phase 2 for acculturation, face and content validity (N=30)

S5 Table: Socio-demographic characteristics of the participants for phase 3 on pilot study (N=40)

S6 Table: Socio-demographic characteristics of the participants for phase 4 to assess the test-retest reliability (N=30)

S7 Table: Sociodemographic characteristics of the participants selected in phase 5 of study on internal consistency of DSEQ. (N=400)

The relevant text has been added in the corresponding sections. 

3. Please ensure you have provided sufficient details to replicate the analyses such as: 

a) a description of any inclusion/exclusion criteria that were applied to participant inclusion in the analysis, 

b) a table of relevant demographic details, 

c) a statement as to whether your sample can be considered representative of a larger population. 

In addition, please state what type of consent was obtained from the parents of children included in this study (for instance, written or verbal, and if verbal, how it was documented and witnessed). We have answered the following queries in the manuscript

a) inclusion/exclusion criteria that were applied to include participants in the study is now incorporated separately for all phases in the methodology section. 

b) Following tables on demographics are added:

S4 Table: Socio-demographic characteristics of the participants for phase 2 for acculturation, face and content validity (N=30)

S5 Table: Socio-demographic characteristics of the participants for phase 3 on pilot study (N=40)

S6 Table: Socio-demographic characteristics of the participants for phase 4 to assess the test-retest reliability (N=30)

S7 Table: Sociodemographic characteristics of the participants selected in phase 5 of study on internal consistency of DSEQ. (N=400)

c) We have incorporated a line regarding the generalizability of the tool in the discussion 

‘the results of this study can be generalized to an area with urban and rural settings, as the last phase of the study was conducted in representative sample size (N=400) from Chandigarh a North Indian Union Territory with both the settings.’ 

We had obtained written informed consent from the participants which in now mentioned in the methods under ethical consideration in the manuscript. 

Page 7, 9, 10, 12

Page 22, line 521-524

Page 13, line 296-301

---

## [Decision Letter · Decision Letter 1]

8 Mar 2021

PONE-D-20-22585R1

Development and Evaluation of the Digital-Screen Exposure Questionnaire (DSEQ) for Young Children

PLOS ONE

Dear Dr. Gupta,

Thank you for submitting your manuscript to PLOS ONE. After careful consideration, we feel that it has merit but does not fully meet PLOS ONE’s publication criteria as it currently stands. Therefore, we invite you to submit a revised version of the manuscript that addresses the points raised during the review process.

We look forward to receiving your revised manuscript.

Kind regards,

Adewale L. Oyeyemi, Ph.D

Academic Editor

PLOS ONE

Reviewers' comments:

Reviewer's Responses to Questions

**Comments to the Author**

1. If the authors have adequately addressed your comments raised in a previous round of review and you feel that this manuscript is now acceptable for publication, you may indicate that here to bypass the “Comments to the Author” section, enter your conflict of interest statement in the “Confidential to Editor” section, and submit your "Accept" recommendation.

Reviewer #1: (No Response)

2. Is the manuscript technically sound, and do the data support the conclusions?

Reviewer #1: Partly

3. Has the statistical analysis been performed appropriately and rigorously? 

Reviewer #1: (No Response)

4. Have the authors made all data underlying the findings in their manuscript fully available?

Reviewer #1: Yes

5. Is the manuscript presented in an intelligible fashion and written in standard English?

Reviewer #1: Yes

6. Review Comments to the Author

Reviewer #1: The statistics for internal consistency and validity are adequate and appear to partly support the investigators' conclusions. The manuscript is basically descriptive with qualitative and some quantitative conclusions.

However, the investigators make statements such as "Parental intervention, media-literacy, and behaviors were the prime correlates that may directly or indirectly modify the home-media environment, activities performed by the child, media-behaviors, and content watched by their children." This modification possibility is an important consideration. It shows that the authors were incomplete in there analysis. There are 3 tables and 7 supplementary tables with many variables and no effort to coordinate this information in a multivariate format. Why was there no consideration of a structure equation modeling (SEM) approach to examine the relationship among underlying structures in the data?

7. PLOS authors have the option to publish the peer review history of their article (what does this mean?). If published, this will include your full peer review and any attached files.

Reviewer #1: No

---

## [Author Response · Author response to Decision Letter 1]

21 Apr 2021

Sr. No. Reviewers' comments Response Page & Line No.

1. The statistics for internal consistency and validity are adequate and appear to partly support the investigators' conclusions. The manuscript is basically descriptive with qualitative and some quantitative conclusions.

However, the investigators make statements such as "Parental intervention, media-literacy, and behaviors were the prime correlates that may directly or indirectly modify the home-media environment, activities performed by the child, media-behaviors, and content watched by their children." This modification possibility is an important consideration. It shows that the authors were incomplete in there analysis. We agree with the reviewer and have now revised the paragraph as follows:

This paper reports the five inter-related studies that focused on developing and testing the psychometric properties of a DSEQ and its correlates among children two to five years of age. This study’s results supported the content validity, test-retest reliability, and internal consistency of the DSEQ. The validated DSEQ tool will be useful in measuring DSE and its correlates among young children in India, who have the most significant growing telecom network with approximately 875 million users [26].

 Page 20, line 543-548

2. There are 3 tables and 7 supplementary tables with many variables and no effort to coordinate this information in a multivariate format. 

 We have removed Table S1 on Medical Subject Heading (MeSh) words. MeSH words are given in an already published review by the lead author [Kaur et al (2019)] . This reference [1] is also cited in the text.

Reference: Kaur N, Gupta M, Malhi P, Grover S. Screen Time in Under-five Children. Indian Pediatr. 2019;56: 773–788. doi:10.1007/s13312-019-1638-8.

We have now presented the background characteristics of the study participants, in different phases of the study, in a single S3 Table. Supplementary Tables S4, S5, S6, S7 are deleted. We would like to clarify that the study participants in different phases of the study were different, hence we cannot coordinate this information in a multivariate format. In each phase a new set of participants were included.

So now, we have submitted 3 main Tables, 1 Figure, 2 Supplementary Tables and 1 Supplementary File in the manuscript as follows:.

Tables

Table I: Description of the digital screen exposure questionnaire (DSEQ) items and constructs.

Table II: Mean ratings of the content and face validation by experts of the digital screen exposure questionnaire.

Table III: Internal consistency and reliability of questions measuring duration, frequency, patterns, and content of screen-time usage among children

Figure

Fig 1: Phases of designing the digital-screen exposure questionnaire (DSEQ)

Supplementary Material 

S1 Table: Likert-type feedback proforma for assessing the face and content validity of the DSEQ.

S2 Table: Socio-demographic characteristics of the participants for various phases

S1 File: Digital Screen Exposure Questionnaire (DSEQ). 

3. Why was there no consideration of a structure equation modeling (SEM) approach to examine the relationship among underlying structures in the data? As per González-Montesinos & Backhoff, 2010, the Structural Equation Model (SEM) offers to determine technical procedures and criteria for validation of measurement models under two conditions: 

1. Conditional independence, understood as a set of latent factors. For instance, skills, attitudes or perceptions, that influences a group of observed variables measured by the questions that make up a scale. The answers are mutually independent, but it is conditioned and dependent by the latent variable which determines them. 

2. Latent factors can be quantified by a dimensional structure based on a Substantive Theory that postulates the existence of psychological constructs that perform a causal influence on people’s responses to a reactive group. 

Reference: Moreno EMO, De Luna EB, Gómez MDCO, López JE. Structural equations model (SEM) of a questionnaire on the evaluation of intercultural secondary education classrooms. Suma Psicol. 2014;21: 107–115. doi:10.1016/s0121-4381(14)70013-x. 

We wish to clarify that in DSEQ, most of the constructs that we have employed were quantitative in nature and were directly observable and it gave direct outcomes rather than latent ones. Therefore, Structural Equation Model was not employed as per the above stated conditions. Also, we would like to further mention that the method and procedure used by us in the present study is widely accepted and used by major researchers in their respective studies for the purpose of development and validation of the study tools [Reference 3 and 29]. 

This is now included in the discussion.

3. Lubans DR, Lonsdale C, Plotnikoff RC, Smith J, Dally K, Morgan PJ. Development and evaluation of the Motivation to Limit Screen-time Questionnaire (MLSQ) for adolescents. Prev Med (Baltim). 2013;57: 561–566. doi:10.1016/j.ypmed.2013.07.023

29. Grover S, Chakrabarti S, Ghormode D, Dutt A, Kate N, Kulhara P. An Indian adaptation of the involvement evaluation questionnaire: Similarities and differences in assessment of caregiver burden. East Asian Arch Psychiatry. 2011;21: 142–151. Page 23

Line 633-638

---

## [Editor Report · Decision Letter 2]

3 Jun 2021

Development and Evaluation of the Digital-Screen Exposure Questionnaire (DSEQ) for Young Children

PONE-D-20-22585R2

Dear Dr. Gupta,

We’re pleased to inform you that your manuscript has been judged scientifically suitable for publication and will be formally accepted for publication once it meets all outstanding technical requirements.

Kind regards,

Adewale L. Oyeyemi, Ph.D

Academic Editor

PLOS ONE
---

## [Editor Report · Acceptance letter]

11 Jun 2021

PONE-D-20-22585R2 

Development and Evaluation of the Digital-Screen Exposure Questionnaire (DSEQ) for Young Children. 

Dear Dr. Gupta:

I'm pleased to inform you that your manuscript has been deemed suitable for publication in PLOS ONE. Congratulations! Your manuscript is now with our production department. 

Kind regards, 

on behalf of

Dr. Adewale L. Oyeyemi 

Academic Editor

PLOS ONE